# Effect of Oral Administration of Active Peptides of *Pinctada Martensii* on the Repair of Skin Wounds

**DOI:** 10.3390/md17120697

**Published:** 2019-12-12

**Authors:** Faming Yang, Xiaoming Qin, Ting Zhang, Chaohua Zhang, Haisheng Lin

**Affiliations:** 1College of Food Science and Technology, Guangdong Ocean University, Zhanjiang 524088, China; yangfm0123@163.com (F.Y.); ZhTing95@163.com (T.Z.); Zhangch2@139.com (C.Z.); haishenglin@163.com (H.L.); 2National Research and Development Branch Center for Shellfish Processing (Zhanjiang), Zhanjiang 524088, China; 3Guangdong Provincial Key Laboratory of Aquatic Products Processing and Safety, Zhanjiang 524088, China; 4Guangdong Province Engineering Laboratory for Marine Biological Products, Zhanjiang 524088, China; 5Key Laboratory of Advanced Processing of Aquatic Product of Guangdong Higher Education Institution, Zhanjiang 524088, China; 6South China Sea Bio-Resource Exploitation and Utilization Collaborative Innovation Center, Zhanjiang 524088, China

**Keywords:** *Pinctada martensii*, mantle, active peptides, skin, wound healing

## Abstract

Skin wound healing, especially chronic wound healing, is a common challenging clinical problem. It is urgent to broaden the sources of bioactive substances that can safely and efficiently promote skin wound healing. This study aimed to observe the effects of active peptides (APs) of the mantle of *Pinctada martensii* on wound healing. After physicochemical analysis of amino acids and mass spectrometry of APs, the effect of APs on promoting healing was studied through a whole cortex wound model on the back of mice for 18 consecutive days. The results showed that APs consisted of polypeptides with molecular weights in the range 302.17–2936.43 Da. The content of polypeptides containing 2–15 amino acids accounted for 73.87%, and the hydrophobic amino acids accounted for 56.51%. Results of in vitro experimentation showed that mice in APs-L group which were fed a low dose of APs (0.5 g/kg bw) had a shortened epithelialization time due to a shortening inflammatory period (*p* < 0.05). Mechanistically, this relied on its specific ability to promote the proliferation of CD31, FGF and EGF which accelerated the percentage of wound closure. Moreover, the APs-L group mice had enhanced collagen synthesis and increased type III collagen content in their wounds through a TGF-β/Smad signaling pathway (*p* > 0.05). Consequently, scar formation was inhibited and wound healing efficiency was significantly improved. These results show that the APs of *Pinctada martensii* promote dermal wound healing in mice and have tremendous potential for development and utilization in skin wound healing.

## 1. Introduction

Skin, with its biological and cellular multilayer structure, is an important barrier to protect the human body from external injury and pathogenic microorganisms [1,2]. Skin damage may occur for a variety of reasons, including acute trauma, chronic wounds, surgery and also in consequence to hereditary diseases [3]. It is noteworthy that in developed countries, people with traumatic diseases which often damage the skin account for about 1.5% of the total population, and their expenses account for 2% to 4% of all medical expenses [4]. For example, in the United States, 6.5 million patients suffer from chronic wounds, which consumes approximately $25 billion annually for their treatment [5]. Skin wound healing is caused by four continuous and overlapping processes: hemostasis, inflammation, proliferation and remodeling [6]. Although the skin is regenerative, perfect recovery rarely occurs. In particular, when skin healing is interrupted, it is often accompanied by permanent scars which may increase in incidence, thus reducing quality of life significantly [7,8]. Therefore, accelerating skin wound healing is essential for better health economy and improving the quality of life of affected people. 

Marine organisms are more complex and resilient than terrestrial organisms due to the physical and chemical properties of their living environment. This makes them rich in diversity and bestowed with unique physicochemical properties which can be harnessed as advantageous natural products [9]. With broadening research on marine active substances in recent years, many marine-derived drugs have been used in clinical treatments such as cytarabine, adenosine and tilidine [10,11]. However, pharmaceutical companies face major challenges in drug production, cost, research and development [12]. Despite the challenges, marine bioactive peptides, especially small peptides with high activity, specificity and stability have continued to attract research attention compared with drugs with higher cost, less safety and therapeutic delivery problems [13]. Marine-derived active peptides have functionality in wound healing, such as the effect of orally administered *Alaska Pollock* low molecular weight peptide on skin wound healing [14]. Moreover, the marine collagen peptide derived from orally ingested salmon (*Oncorhynchus keta*) can improve wound healing after cesarean section in rats [15], while the sea cucumber low molecular weight oligopeptide has efficacy in diabetic wound healing in mice [16].

*Pinctada martensii* is the main shellfish for growing pearls in coastal areas such as Guangdong and Guangxi in southern China [17]. The *Pinctada martensii* mantle containing cells that secrete nacre plays an important role in the formation of the shell. Following the formation of the pearl, it is implanted into the pearl gonad by the nucleus pulposus [18]. At present, pearl shellfish are often discarded after harvesting pearls, resulting in low realization of their economic potential [19]. The study found that *Pinctada martensii* mother-of-pearl meat is rich in protein and is a valuable source of protein for human nutrition [20]. Therefore, it is of great economic importance to develop processing techniques for converting such underutilized by-products into food sources, particularly considering the potential medical benefits associated with increased incorporation into local diets. Hydrolysates obtained by controlled enzymatic hydrolysis have a variety of biological activities. A wide range of protein hydrolysates are mainly used as nutritional supplements, while custom hydrolysates with a defined composition are usually used in special medical diets [21,22,23]. However, the effect of the mother-of-pearl coat membrane protein peptide of *Pinctada martensii* on wound healing has not been reported.

Therefore, in this study, the membrane active peptides (APs) of *Pinctada martensii* were obtained, their characteristics determined, and their effects in promoting wound healing were evaluated by a full-thickness, mouse skin wound model. The work sought to investigate a theoretical potential for therapeutic product development in subsequent studies.

## 2. Results 

### 2.1. Basic Composition Analysis of APs 

Discovered through previous research [23], the molecular weight range of APs was 302.17–2936.43 Da; among the 17 amino acids of APs were glutamic acid (5.66 g/100 g), alanine (2.71 g/100 g), glycine (2.015 g/100 g), leucine (2.07 g/100 g) and phenylalanine (1.70 g/100 g). Essential amino acids (excluding tryptophan) accounted for 33.77% of the total amino acids. Among them, essential amino acids such as glutamic acid and proline (except tryptophan), leucine and isoleucine (branched chain amino acid) accounted for 21.83% and 18.89% of the total amino acids, respectively. Peptide fingerprinting of 21 characteristic peptides in APs was performed using liquid chromatography-mass spectrometry (LC-MS/MS). The molecular weights of APs were in the range 1512.83 Da to 2241.05 Da (amino acid residue 10–21). Two peptide fragments of Gln-Leu and Asp-Leu recurred in the 21 characteristic peptide sequences. Leu recurred at the beginning of the characteristic peptide, while Asn and Arg reappeared at the tail of the characteristic peptide. Additionally, Val, Leu, and Ile were found in the middle of the APs characteristic peptides.

### 2.2. The Effect of APs on Wound Healing

Figure 1 and Table 1 show the effect of orally ingested APs on skin percentage of wound closure and epithelialization time in mice. Figure 1A shows representative photos of the animals in each experimental group during and after wound healing. As shown in Figure 1B, there was no significant difference in percentage of wound closure between the experimental groups in the first 4 days after modeling (*p* > 0.05). Compared with the other groups, the positive control group and the APs-L group significantly promoted wound closure at 6 days and 10 days after modeling (*p* < 0.05). On day 12, the positive control and each of the APs-administered groups had significantly accelerated skin wound healing, compared with the negative control group (*p* < 0.05). On the 14th day, there was no significant difference in wound healing between the experimental groups (*p* > 0.05), but the healing extent of the negative control group was only 90%, while the positive control group, APs-L and APs-H were 96%, 96% and 98%, respectively, nearly completely healed, maybe as a matter of individual differences. However, by the 16th and 18th days, there was no significant difference in wound healing between the experimental groups (*p* > 0.05), and the wounds were completely closed.

During the epithelialization process, the blood and tissue fluid produced by the wound and the tissue repair gradually condensed to form black scabs. As can be seen from Table 1, the APs-L group (8.5 ± 1.29 days, 13.0 ± 1.15 days, respectively) significantly promoted the shedding and epithelialization of the black wound sputum compared with the negative control group (10.5 ± 0.58 days, 15.3 ± 1.89 days, respectively) (*p* < 0.05), which is consistent with the results of Figure 1.

### 2.3. Effects of APs on Wound Cytokines

As can be seen from Figure 2A, on day 3, the IL-1β content of the APs-L group was relatively lower compared with the negative control group, while the IL-1β content of the APs-H was higher than that of the negative control group. By the fifth day, the IL-1β content of the APs-L group was the lowest, but not significantly (*p* > 0.05). As shown in Figure 2B, on the third day, the IL-10 contents of the APs-L and APs-H groups were significantly higher than that of the negative control group (*p* < 0.05). On day 5, the IL-10 contents of the APs-L and APs-H groups were lower than that of the negative control group, but there was no significant difference among experimental groups (*p* > 0.05).

### 2.4. Effects of APs on the TGF-β/Smad Signaling Pathway

To investigate the mechanism of action of APs on TGF-β/Smad signaling pathway, we further identified APs for transforming growth factor-β1 (TGF-β1), transforming growth factor-β receptor II (TβRII) and Smad7 in the TGF-β/Smad signaling pathway. As shown in Figure 3A, the content of TGF-β1 in the positive control and the APs-L groups was significantly higher than that in the other groups (*p* < 0.05). Figure 3B shows that the levels of TβRII in APs-L and APs-H were 4.91 ng/μg protein and 4.41 ng/μg protein, which was not significantly different from the negative control group (3.83 ng/μg prot) (*p > 0.05*). However, Aps-L and Aps-H increased the expression of TβRII compared to the negative control group. From Figure 3C, the Smad7 content of the APs-L and APs-H groups was significantly lower than that of the negative control group (*p* < 0.05). 

### 2.5. Histological Evaluation of Mouse Wounds Treated with APs

The effect of APs on skin wound healing in mice was studied microscopically using H&E staining in Figure 4. On day 3 of the experiment, inflammatory cells infiltrated the wounds in the negative control group. However, the positive control group, APs-L and APs-H groups showed a weaker inflammatory response. On day 7 of the experiment, there were still observable inflammatory cells in the negative control group. The epidermal layer of the wound was incompletely healed and the collagen fibers in the dermis were sparse and few. In contrast, in the positive control group, APs-L and APs-H groups, no inflammatory reaction was observed. Moreover, the wounds formed a coherent epidermis, and more granulation tissue formation in the dermis. Compared with the negative control group on day 18 of the experiment, the repair of epidermis and dermis in the positive control, APs-L and APs-H groups resembled normal skin.

### 2.6. APs Promote the Expression of CD31, FGF and EGF

As shown in Figure 5, on days 7 and 18, there was no significant difference in the effects of APs-L and APs-H on CD31 and FGF compared with the positive control group (*p* > 0.05), but the expression level was higher than that of the negative control group. On day 7, compared with the negative control group, the APs-L group of mice significantly had promoted production of EGF (*p* < 0.05); on the 18th day, the positive control group significantly promoted the production of EGF (*p* < 0.05), and there was no significant difference in the other experimental groups. Therefore, APs promoted the expression of CD31, FGF, and EGF compared with the control group.

### 2.7. Effect of APs on Collagen and Scar Formation

In this experiment, a polarized light microscope was used. Under the microscope, the type I collagen fiber was yellowish red, and the type III fiber was green. Figure 6A shows a large number of tightly arranged collagen fibers in the APs administration groups, showing a strong birefringence, a yellow or red type I fiber; showing a weak birefringence and a green type III fiber. As can be seen in Figure 6B, the collagen I/III ratio in the administration groups was small, and the difference was significant (*p* < 0.05). The scar reduction rate of mice at 18 days after wounding is as shown in Figure 6C. The scar reduction rate of the APs-L group was significantly higher than that of the negative control group (*p* < 0.05). 

## 3. Materials and Methods 

### 3.1. Materials

In May 2019, *Pinctada martensii* meat was purchased from Dongfeng Market, Zhanjiang City, China. The Chuan I type product was purchased from China Jiangzhong Group; SPF male mice weighing 20 ± 2 g were purchased from Pengyue Experimental Animal Breeding Co., Ltd. (Jinan, China) and the production license number was SCXK 20190003, Shandong, China. The laboratory license number was SYXK 2019-0204, Guangdong, China. Neutral Protease (3 × 10^5^ U/g) was purchased from Pangbo Biological Engineering Co., Ltd. (Nanning, China) The other chemicals used in this experiment were of analytical grade and are commercially available.

### 3.2. Preparation of APs

The mother-of-pearl mantle of *Pinctada martensii* was washed, drained, and water was added to the ratio of 1:3 (mantle:water), followed by the addition of neutral protease at 1000 U/g (raw material). The hydrolysis reaction was performed at 53 °C for 5 h. Subsequently, the enzyme was inactivated at 100 °C for 10 mins followed by centrifugation at 8000 rpm at 4 °C to obtain the supernatant (Thermo Sorvall LYNX6000, Thermo, Waltham, MA, USA). The macromolecules and particulate impurities were removed through a 200 μm ceramic membrane microfiltration device (WTM-1812G, Hangzhou Woteng Membrane Engineering Co., Ltd., China), and the APs were obtained through the <3 kDa ultrafiltration system (Mini Pellicon, Millipore, USA). The samples were freeze-dried to prepare a lyophilized powder for future experimentation (FD-551, EYELA, Tokyo, Japan).

### 3.3. Determination of Molecular Weight Distribution

APs were subjected to reductive alkylation treatment: 100 μL/tube of APs were mixed with 10 mmol/L dithiothreitol in a water bath at 56 °C for 1 h. Then, 55 mmol/L iodoacetamide was added to the 100 μL/tube and the reaction was carried out in the dark at room temperature for 1 h. Next, the mixture was desalted using a self-priming desalting column and the solvent was evaporated in a vacuum centrifuge at 45 °C. The LTQ VELOS ESI cation calibration solution was used for molecular weight distribution analysis by liquid chromatography-mass spectrometry (LC-MS/MS). The formulation was: caffeine (2 μg/mL), MRFA (1 μg/mL), Ultramark 1621 (0.001 %), and n-butylamine (0.0005%) in acetonitrile (50%), aqueous methanol (25%) and acetic acid (1%). 

### 3.4. Amino Acid Composition of APs

The APs were hydrolyzed using 6 mol/L HCl. The solution was analyzed with an amino acid analyzer (L-8900, Hitachi, Tokyo, Japan). 

### 3.5. Major Peptide Sequence Analysis of APs

APs were identified using Thermo Q Exactive™ Hybrid Quadrupole-Orbitrap™ Mass Spectrometer with electrospray ionization (ESI) interface (Thermo Fisher Scientif, USA). The secondary mass spectrometry was based on the results of APs primary mass spectrometer total ion map. APs analysis was performed on a mass spectrometer set from 350 to 1800 m/z. The mass spectrometer was operated under the following conditions: Resolution: 75,000, AGC target: 1e5, Maximum IT: 60ms, TopN: 20, NCE/steppedNCE: 27.

### 3.6. Mouse Experiments and Wound Model

The mice were randomly divided into three groups, and maintained on standard rodent chow and water ad libitum, 19 mice per group, including a negative control group (dose: 0.1 mL/10 g bw saline), positive control group (Chuyan I type product, dose: 0.1 mL/10 g bw), APs-L (0.5 g/kg bw) and APs-H (2 g/kg bw) group. After the mice were anesthetized with 1% pentobarbital sodium (50 mg/kg), their back was shaved and disinfected, and the whole layer of skin with a diameter of 0.8 cm was cut off creating a full cortical wound animal model. The mice were housed in cages and gavaged with APs daily. The wounds were observed and photographed every two days. Each set of experiments was performed in triplicates. The study was approved by the Guangdong Ocean University (Zhanjiang, China) Experimental Animal Care Ethics Committee (Approval No.: 20190001, Approval Date: 17 June 2019). All experiments were in accordance with the ARRIVE guidelines and were conducted in accordance with the National Institutes of Health guidelines for the care and use of laboratory animals (NIH Publication 8023, revised 1978). The mice were sacrificed on the third, fifth, seventh, and eighteenth days. The tissues surrounding the wounds of the mice were taken for measurement of inflammatory factors, cytokines, H&E staining for microscopy and immunohistochemistry. Sirius red picric acid staining was also performed at the wound site of the mice.

### 3.7. Percentage of Wound Closure and Residual Scar Rate

The wound area was measured by tracking the edge of the wound using Image J image analysis software. The percentage of wound closure was expressed as the ratio of the remaining area of the wound to the original area of the wound over time. The percentage of residual scar rate was expressed by taking the ratio of the scar area to the original area of the wound. 

### 3.8. Tissue Preparation for Histological Assessment

Tissue removed from the wound site of each mouse was immersed in neutral formalin for fixation, then dehydrated with a series of increasing alcohol concentrations. Next, the tissue was embedded in paraffin, and finally sliced for histopathology and histomorphological observation.

### 3.9. ELISA Analysis

Preparation of mouse wound tissue homogenate: The modeled mice were sacrificed by cervical dislocation and the normal tissues of the wound and the wound of 2.0 mm were created. According to the ratio of 1:9 (wound tissue: PBS), PBS was added, and the electric grinder was homogenized at 4 °C to prepare a homogenate. The supernatant of homogenized tissue was collected for protein extraction by centrifuging at 10,000 r/min for 15 min at 4 °C. The quantitative analysis of IL-1β (interleukin-1β), TGF-β1 (transforming growth factor β), TβRII (transforming growth factor beta 1 type II receptor), Smad7 (signal transduction molecule 7, antibody from Mlbio, Shanghai, China), and IL-10 (interleukin-10, antibody form Nanjing Institute of Bioengineering, Nanjing, China) in 10% (*v*/*v*) dorsal skin homogenate supernatant was conducted using ELISA kits. The specific measurement steps were performed according to the kit instructions. 

### 3.10. Hematoxylin and Eosin (H&E) Staining for Microscopic Analysis

After the mice were sacrificed, the tissues were taken for histological analysis. The samples were fixed in 4% paraformaldehyde at 4 °C for 24 h, followed by dehydration by immersing in a 70–100% ethanol solution. The staining procedure involved staining the skin tissues of mice with H&E staining prior to performing histopathological analysis using a Japanese Olympus microscope (Olympus IX51, Tokyo, Japan). 

### 3.11. Immunohistochemistry

We used immunohistochemical techniques to evaluate the expression of EGF (epidermal growth factor), FGF (fibroblast growth factor), and CD31 (platelet endothelial cell adhesion molecule-1, PECAM-1). After the tissue sections were dewaxed, hydrated and the hydrothermal antigen was repaired, the enzyme was inactivated using a 3% H_2_O_2_-methanol solution, and the antigen was recovered with a citrate buffer (pH 6.0). Subsequently, 50–100 µL of ready-to-use goat serum was added dropwise to block non-specific binding sites, followed by incubation at room temperature for 20 mins. Next, 50–100 µL of each primary antibody (diluted 1:200, courtesy of Nanjing Institute of Bioengineering, Nanjing, China) was added dropwise to the tissue slices, and the slides were incubated at 37 °C for 2 h in a humidified chamber. A universal secondary IgG antibody-Fab fragment-HRP multimer (50 µL, courtesy of Nanjing Institute of Bioengineering, Nanjing, China) was added dropwise, and the slides were incubated at 37 °C for 30 mins at room temperature, then washed 3 times with PBS. After color development with diaminobenzidine (DAB), the slides were evaluated under an optical microscope (Olympus IX51, Japanese), and each photo was analyzed using Image-pro plus 6.0 (Media Cyber netics, Inc., Rockville, MD, USA) software to obtain positive cumulative light density.

### 3.12. Sirius Red Picric Acid Dyeing

The prepared sections were subjected to conventional dewaxing. Then they were stained with Sirius red picric acid staining solution for 8–10 min. Running water was used to remove residual staining from the surface of the slides. The sections were then dehydrated with absolute ethanol and sealed with neutral gum. Samples stained with sirius red picric acid solution were examined by polarized light microscopy (magnification: ×200), and collagen (type I and type III) in granulation tissue was quantitatively analyzed using Image-Pro plus 6.0 (Media Cybernetics, Inc., Rockville, MD, USA).

### 3.13. Data Analysis

Experimental data are expressed as mean ± standard deviation (mean ± SD). Statistical analysis was performed using SPSS software (version 20). Multiple comparisons between groups were performed by the LSD method. The significance level was set to *p* < 0.05 and the extremely significant level was set to *p* < 0.01. Experimental images were processed using Image J to calculate the wound area.

## 4. Discussion

Previous studies had found that topical use of APs could promote the healing of mouse skin wounds [23]. However, the mechanism of action of oral and topical APs is different, so we need to explore the mechanism of oral APs on wound healing. 

Through the use of the mouse skin full cortical wound model, we demonstrated the effect of gastric administration of APs on wound healing and related mechanisms. Nutrition is an important part of wound healing, and maintaining good nutrition and adequate levels of nutrient cycling is essential for overall health and wound healing [24]. In this study, essential amino acids (excluding tryptophan) accounted for 33.77% of the total amino acids in hydrolyzed *Pinctada martensii*, which provides excellent nutritional support for wound healing. Moreover, the hydrophobic amino acids closely related to the immunological activity accounted for 56.51% of the total amino acids [25]. This provided a theoretical basis for APs to improve immunity and produce inflammatory reactions in the inflammatory phase of skin wound healing. At the same time, APs rich in glutamate, glycine, cysteine and phenylalanine can achieve nitrogen balance during wound healing, which is beneficial for angiogenesis, fibroblast proliferation, collagen synthesis and wound weight [26]. In terms of nutrition, our findings provide a theoretical basis for the wound healing potential of APs.

Marine organisms are a rich source of structurally diverse biologically active compounds with a variety of biological activities [27]. Peptide sequences comprising Gln-Leu and Asp-Leu were repeatedly present in the APs of *Pinctada martensii* functional peptide. Studies have shown that Gln-Leu and Asp-Leu play important roles in the removal of free radicals from peptides [28,29,30]. And Asp often appears in the middle of the characteristic peptide. Another study found that arginine, a non-essential amino acid, is an essential matrix component for some stressed adults. Therefore, Glu-Leu and Ala-Arg play important roles in active peptides that promote wound healing [31]. Other studies have also shown that arginine supplementation enhances collagen deposition [32,33]. In addition, small molecular weight (< 3000 Da) APs can be quickly absorbed and provide a nutrient matrix for wound healing. 

After injury, the inflammatory response is both normal and critical in restoring tissue homeostasis [12]. Within three to five days after injury, macrophages are the most prominent cells in the healing tissue [34]. Generally, macrophages can be divided into two phenotypes: the M1 phenotype (secreting IL-1β, TNF-α, etc.) and the M2 phenotype (secreting IL-10, IL-13, etc.) [35,36]. The function of M1 macrophages is to remove pathogenic microorganisms and wound debris from injured tissues, which promotes a strong inflammatory response through the secretion of pro-inflammatory factors. M2 macrophages promote tissue repair, inflammatory regression and immune regulation. Wounds infiltrated by macrophages mainly showed M1 phenotype in early stages [37]. Although APs could not significantly reduce IL-1β content, mice in the APs-L group had lower IL-1β content than those in the negative control group (Figure 2); this may inhibit the subsequent activation of skin-resident immune cells [38]. The major macrophage population in the wound on day 5 was the M2 macrophage population [39]. In normal wounds, M2 macrophages may occupy the main site of the wound early in the healing process, and the inflammation period is very short due to the lack of bacterial products or excessive dead tissue material [34]. On the third day, compared with the negative control group, the positive control group, APs-L and APs-H groups had significantly promoted secretion of IL10 (Figure 2), which was conducive to inhibiting the inflammatory response and shortening the inflammatory phase, consistent with the results of H&E (Figure 4) [23].

The proliferative phase is characterized by three main processes: re-epithelialization, neovascularization and granulation tissue formation [40]. During the proliferative phase, fibroblasts migrate from the surrounding tissue to the wound area, and collagen synthesis begins and proliferates to generate new granulation tissue [41]. Expression of CD31 in endothelial cells can be used to indicate the level of angiogenesis in wound tissue [42]. Corneal-forming cells (EGF) and fibroblasts (FGF) play crucial roles in wound healing [43]. Moreover, the wound healing process requires structural and functional reconstruction as well as contraction of differentiated muscle fibroblasts [44]. In this study, mice in the APs-L group had significantly promoted EGF secretion on day 7, but it can also be seen that they also had a certain effect on the proliferation of CD31 and FGF compared with the negative control group (Figure 5), which was different from the results of immunohistochemistry of topical APs. This may be caused by different mechanisms of action [23], or may also be because the healing of wounds in mice rather than humans is mainly achieved by contraction rather than epithelial re-epithelialization and granulation tissue formation, resulting in differences in action [45]. This accelerated skin wound healing in mice, consistent with the results of the healing rate, epithelialization time (Figure 1 and Table 1) and H&E (Figure 4). There was no significant difference in CD31, EGF and FGF in the APs-H group compared with the negative control group.

Remodeling is the longest period of wound healing. Wound contraction is a necessary process to optimize healing [46]. It involves replacing the major collagen found in connective tissue with stronger type I collagen to replace type III produced during the proliferative phase. The level of collagen III is significantly increased after the inflammatory phase and can reduce scar formation [37]. From the results of H&E on day 18, the APs-L group had significantly enhanced wound epithelialization and granulation tissue formation (Figure 4), which is consistent with the results of Figure 1. There was no significant difference between APs-H and the negative control group. At the same time, compared with the negative control group, the APs-L group experienced an effect on FGF, which thereby promoted the secretion of collagen and accelerated wound healing (Figure 5) [23]. Our studies showed that the APs-L group could influence d collagen synthesis through the TGF-β/Smad signaling pathway and thus assist wound healing (Figure 3). Furthermore, the APs shortened the inflammatory phase, significantly promoted secretion of TGF-β1 and appropriately promoted the content of type III collagen, thereby inhibiting scar formation in APs-L treated mice as demonstrated by the results summarized in Figure 1 and Figure 6C. There was no significant difference between APs-H and the negative control group.

## 5. Conclusions

In this study, APs extracted from the mantle of *Pinctada martensii* consisted of a polypeptide with a molecular weight of less than 3 kDa and had a characteristic sequence. *In vivo* tests have shown that mice in the APs-L group shortened the epithelialization time by inhibiting inflammatory response. In addition, they attained the ability to promote the proliferation of CD31, FGF and EGF, and improve the percentage of wound closure, thereby promoting wound healing. Finally, APs-L group mice can affect collagen synthesis through the TGF-β/Smad signaling pathway, and the collagen I/III ratio is appropriate, inhibiting scar formation and improving healing quality. Future research will explore specific molecular mechanisms involved in wound healing mediated by *Pinctada martensii*.

## Figures and Tables

**Figure 1 marinedrugs-17-00697-f001:**
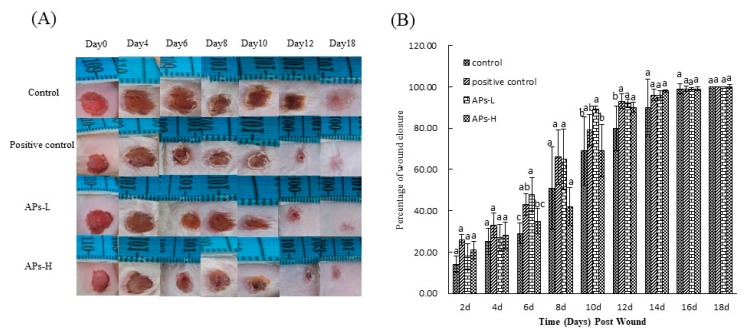
Skin wound healing over time in mice treated with *Pinctada martensii* active peptides (APs). (**A**) Representative photographs of the wounds in excision wound model. (**B**) Percentage of wound closure obtained on days 2, 4, 6, 8, 10, 12, 16 and 18 from vehicle or APs treated mice. Values are expressed as mean ± SD, n = 4. Note: Different superscript letters on the same day indicate significant differences between the groups (*p* < 0.05) and insignificant differences (*p* > 0.05), respectively.

**Figure 2 marinedrugs-17-00697-f002:**
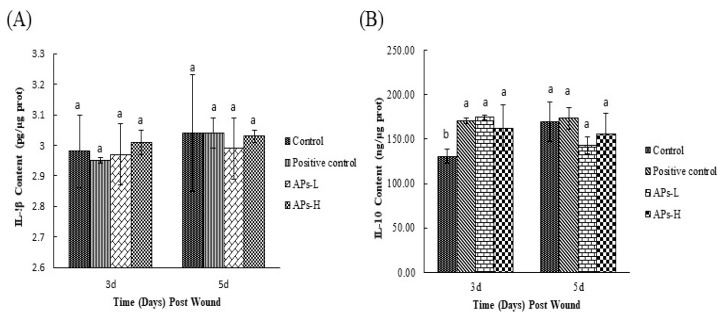
Effect of *Pinctada martensii* APs on cytokines in wound skin tissue. (**A**) Effect of APs-L and APs-H on the expression of IL-1β. (**B**) Effect of APs-L and APs-H on the expression of IL-10. Note: Different superscript letters on the same day indicate significant differences between the groups (*p* < 0.05) and insignificant differences (*p* > 0.05), respectively.

**Figure 3 marinedrugs-17-00697-f003:**
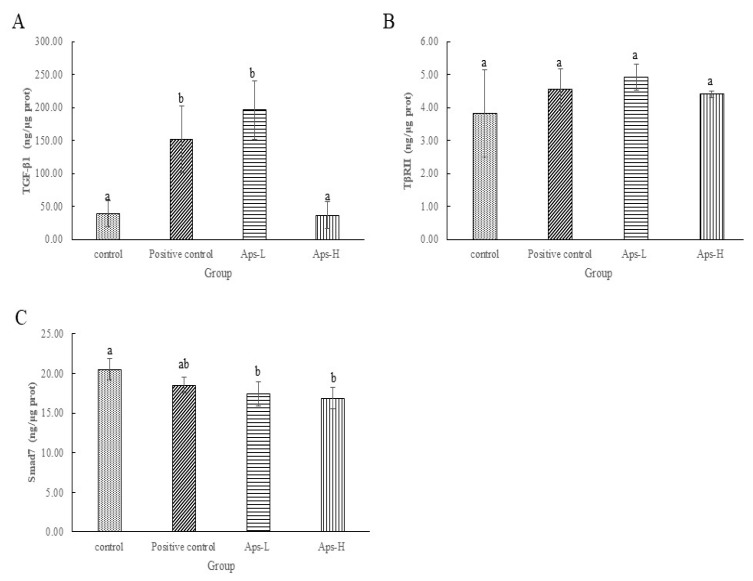
Effects of *Pinctada martensii* APs on the TGF-β/Smad signaling pathway. (**A**) Effect of APs-L and APs-H on the expression of TGF-β1. (**B**) Effect of APs-L and APs-H on the expression of TβRII. (C) Effect of APs-L and APs-H on the expression of Smad7. Note: Different superscript letters on the same day indicate significant differences between the groups (*p* < 0.05) and insignificant differences (*p* > 0.05), respectively.

**Figure 4 marinedrugs-17-00697-f004:**
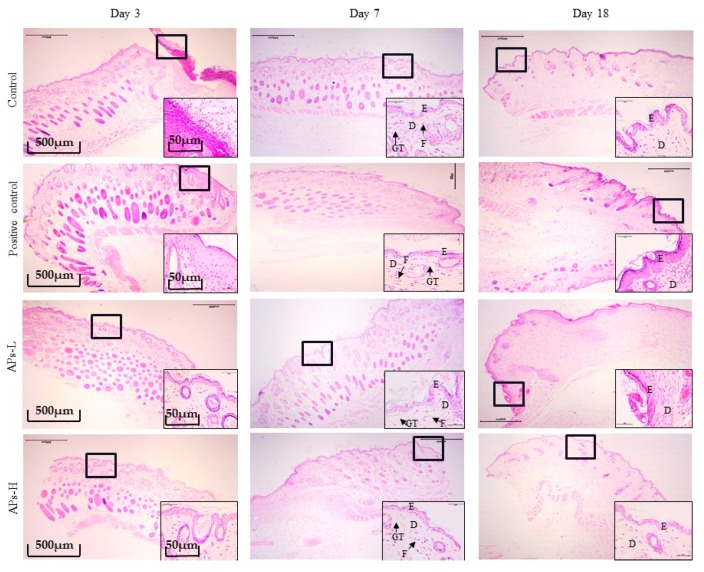
H&E stain histological analysis of mouse wounds treated with *Pinctada martensii* APs (4× magnification). Note: The black bold frame indicates the wound site, and the picture at the bottom corner of the picture shows a magnified image of the wound (40×). On the third day, black arrows indicate inflammatory cell infiltration. The letters D, E, F, and GT represent the dermis layer, the epidermal layer, fibroblasts, and granulation tissue, respectively.

**Figure 5 marinedrugs-17-00697-f005:**
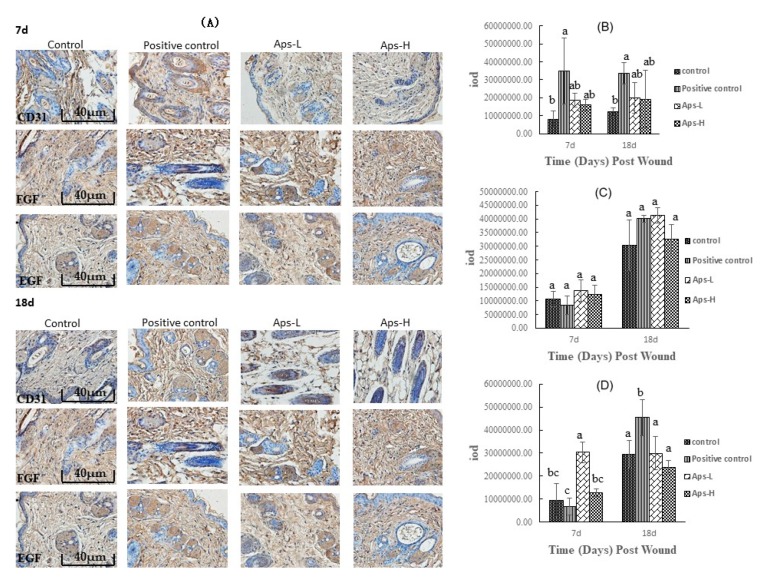
Immunohistochemical analysis of mouse skin wounds treated with *Pinctada martensii* APs. (**A**) Representative images of CD31, FGF and EGF immunostaining on days 7 and 18 (scale bar: 40 μm). (**B**) Expression of CD31 at 7d and 18d after trauma. (**C**) Expression of FGF at 7d and 18d after trauma. (**D**) Expression of EGF at 7d and 18d after trauma. Note: Different superscript letters on the same day indicates significant differences between the groups (*p* < 0.05) and insignificant differences (*p > 0.05*), respectively.

**Figure 6 marinedrugs-17-00697-f006:**
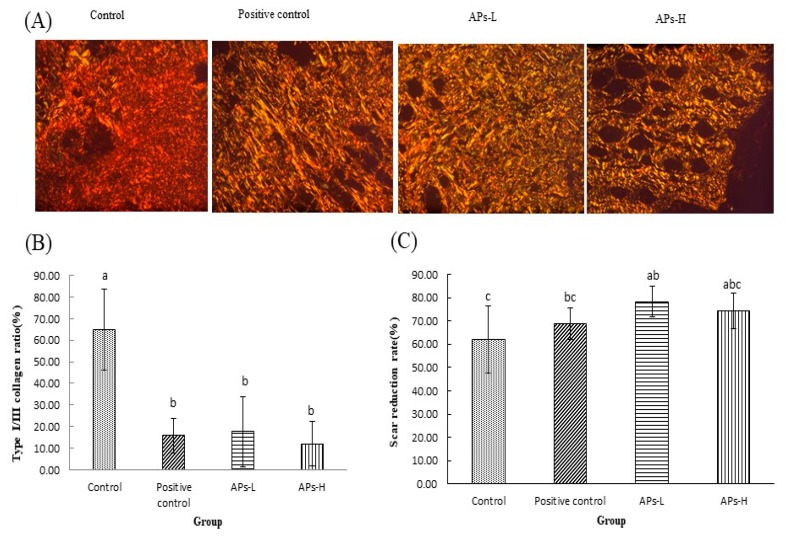
The effect of *Pinctada martensii* APs on skin scar formation and collagen composition in wounded mice. (**A**) On day 18, representative images of Sirius red staining of sections of the control, APs-L and APs-H groups on day 18 (magnification: ×200). Collagen I appears to be yellowish red, while collagen III appears to be green. (**B**) Expression of collagen I/III at 18d after trauma. (**C**) Quantitative analysis of scar area of skin wounds on the 18th day after modeling. Note: Different superscript letters on the same day indicates significant differences between the groups (*p* < 0.05) and insignificant differences (*p* > 0.05), respectively.

**Table 1 marinedrugs-17-00697-t001:** Effect of *Pinctada martensii* APs on skin wound healing time in mice.

Group	Dislocation Time (Day)	Epithelialization Time (Day)
Control group	10.5 ± 0.58 ^bc^	15.3 ± 1.89 ^a^
Positive control group	9.5 ± 1.00 ^cd^	13.8 ± 0.50 ^ab^
APs-L group APs-H group	8.5 ± 1.29 ^d^ 10.8 ± 1.26 ^bc^	13.0 ± 1.15 ^b^ 14.3 ± 0.50 ^ab^

Note: Dislocation time refers to the time when the black sputum (as shown in Figure 2A) of the wound surface is completely removed, as one of the indicators in the process of epithelialization. Different superscript letters on the same day indicate significant differences between the groups (*p* < 0.05) and insignificant differences (*p* > 0.05), respectively.

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
