# Peer review of "Effect of Oral Administration of Active Peptides of Pinctada Martensii on the Repair of Skin Wounds"

_marinedrugs, 2019, doi:10.3390/md17120697_

Round 1

Reviewer 1 Report

The suthors perfomed interesting investigations regarding the effect of oral administration of "Active Peptides" (What does this expression mean? Are others inactive, and why?). on the repair of skin wounds.

There are 3 points to be considered:

1) Although the suthors performed intensive mass spectrometry for the analysis  of the peptides, they give only a rough overview on the constituing amino acids of the peptides. In particular, a sequence analysis would bring much better insights. Why was it not possible to provide such data?

2) Very interesting is the fact that an oral administration leads to such therapeutic effects. A prerequisite for this is the resorption of the peptide in the intestine. I am not aware that such resorption has ever been observed for peptides. Do the authors think that such resorption takes place, in particular regarding the low pH and the presence of proteases in the stomach? 

3) Citations are incomplete: At least three papers of Pfalzgraff et al must be cited; Frontiers Pharmacol 9(2018); Sci. Rep. 6, 2016; Br. J.Pharmacol. 175 (2018).   

Furthermore, the bars in Fig. 1B are not convincing, maybe the illusztration can be improved. 

Author Response

Dear professor,

We are very grateful to you for your detailed review of our imperfect papers, and for your critical suggestions. We must admit that we have a lot of unclearness about the method of introduction of the article, which has caused you too many obstacles in reviewing the manuscript. The following is our reply based on the comments you gave:

Question 1: The suthors perfomed interesting investigations regarding the effect of oral administration of "Active Peptides" (What does this expression mean? Are others inactive, and why?). on the repair of skin wounds.

Response 1: The active peptides in the title means that we had prepared peptides that promoted wound healing of mouse skin from Pinctada Martensii, which means that our peptides had functional activity to promote wound healing in mice, so it is defined as: active peptides of Pinctada Martensii.

However, this does not mean that we deny that the peptides of other studies are inactive. At the same time, the use of active peptides names is not our first. There was referenced to the naming methods of the titles of their articles, for example: "Designing improved active peptides by Gomes et al." for therapeutic approaches against infectious diseases "; "Orally Active Peptides: Is There a Magic Bullet? " by Andreas F. B. et al.

If you think that it would be more scientific and reasonable to modify the existing naming, we will modify it based on your suggestions.

Question 2: Although the suthors performed intensive mass spectrometry for the analysis of the peptides, they give only a rough overview on the constituing amino acids of the peptides. In particular, a sequence analysis would bring much better insights. Why was it not possible to provide such data?

Response 2: Thank you for your question, this is also our tangled issue. Because as the same sample, in the previous research, we performed a study of topical administration on skin wound healing in mice and published the results: Evaluation of Small Molecular Polypeptides from The Mantle of Pinctada Martensii on Promoting Skin Wound Healing in Mice.

In the published article, we had used the mass spectrometry and amino acid data of the sample, and the results had been analyzed in detail, but considering the problem of data reuse, we decided to summarize the main results here. We also hope that you can help us solve this problem, and please give us more valuable suggestions.

Question 3: Very interesting is the fact that an oral administration leads to such therapeutic effects. A prerequisite for this is the resorption of the peptide in the intestine. I am not aware that such resorption has ever been observed for peptides. Do the authors think that such resorption takes place, in particular regarding the low pH and the presence of proteases in the stomach?

Response 3: The question you raised is very important. Regarding the content of your research, we have not yet carried out, but according to our inference that such absorption had occurred, this will be the content of our subsequent research and the relevant results will be carried out at the appropriate time post.

Question 4: Citations are incomplete: At least three papers of Pfalzgraff et al must be cited; Frontiers Pharmacol 9(2018); Sci. Rep. 6, 2016; Br. J.Pharmacol. 175 (2018).  

Response 4: Thanks for your suggestion, we have cited the references in the appropriate place in the text.

Question 5: Furthermore, the bars in Fig. 1B are not convincing, maybe the illusztration can be improved.

 Response 5: The shortcomings you pointed out are also regrettable in our results. At the same time, the results of this part are so carefully thought out. During the experiment, we had 6 mice in each group. When we processed the data, we took into account the issue of individual differences, so we scientifically selected the data of 4 mice for analysis. Because if the amount of data is too small, it will be more difficult to convince our research results, of course, we are also improving the results analysis section.

Finally, we thank you again for your valuable suggestions for our work!

Kind regards,

Faming Yang

December 5, 2019

Reviewer 2 Report

The ms needs only few modifications before acceptance. In particular, I suggest to prepare a final figure with the details developed in the manuscript. Moreover, as major concerns, I suggest to test the peptide on an in vitro model of scratch wound assay in order to assess the potency and efficacy directly on cell lines.

Author Response

Dear professor,

We are very grateful to you for your detailed review of our imperfect papers, and for your critical suggestions and shortcomings. We must admit that we have a lot of unclearness about the method of introduction of the article, which has caused you too many obstacles in reviewing the manuscript. The following will be my response based on the comments you gave:

Question 1: The ms needs only few modifications before acceptance. In particular, I suggest to prepare a final figure with the details developed in the manuscript. Moreover, as major concerns, I suggest to test the peptide on an in vitro model of scratch wound assay in order to assess the potency and efficacy directly on cell lines.

Response 1: Thank you for your suggestions, we will modify the article based on your suggestions. As for the cell experiment, we are now working on it. At present, we have performed proliferation experiments of HaCaT cells and HSF cells respectively. The peptide concentration was set to 6.25 μg / ml, 12.5 μg / ml, 25 μg / ml, 50 μg / ml, 100 μg / ml, 200 μg / ml, 400 μg / ml. The results showed that HaCaT had the highest proliferation rate when the peptide concentration was 50 μg / ml, and HSF had the highest proliferation rate when the peptide concentration was 100 μg / ml. Cell migration tests will be carried out in subsequent experiments and published, and you will see if you need to increase the results of the cell tests.

Finally, we thank you again for your valuable suggestions for our work!

Kind regards,

Faming Yang

December 5, 2019

Round 2

Reviewer 1 Report

The paper is now ready for publication.